# The Influence Mechanism of Temperature and Storage Period on Polarization Properties of Poly (Vinylidene Fluoride–Trifluoroethylene) Ultrathin Films

**DOI:** 10.3390/membranes11050301

**Published:** 2021-04-21

**Authors:** Xingjia Li, Zhi Shi, Xiuli Zhang, Xiangjian Meng, Zhiqiang Huang, Dandan Zhang

**Affiliations:** 1School of Mathematics, Physics and Statistics, Shanghai University of Engineering Science, Shanghai 201620, China; xjli@sues.edu.cn (X.L.); shsz24@163.com (Z.S.); huangzhiqiang120@gmail.com (Z.H.); zdd1284995601@163.com (D.Z.); 2Research Center for Advanced Mirco-and Nano-Fabrication Materials, Shanghai University of Engineering Science, Shanghai 201620, China; 3Key Laboratory of Infrared Physics, Shanghai Institute of Technical Physics, Chinese Academy of Sciences, Shanghai 200083, China; 4Key Laboratory of Infrared Imaging Materials and Detectors, Shanghai Institute of Technical Physics, Chinese Academy of Sciences, Shanghai 200083, China

**Keywords:** ferroelectric, P(VDF–TrFE) ultrathin films, temporal stability, molecular modeling, polarization switching

## Abstract

The effect of testing temperature and storage period on the polarization fatigue properties of poly (vinylidene fluoride-trifluoroethylene) (P(VDF–TrFE)) ultrathin film devices were investigated. The experimental results show that, even after stored in air for 150 days, the relative remanent polarization ($P_{r}/P_{r}(0)$) of P(VDF–TrFE) of ultrathin films can keep at a relatively high level of 0.80 at 25 °C and 0.70 at 60 °C. To account for this result, a hydrogen fluoride (HF) formation inhibition mechanism was proposed, which correlated the testing temperature and the storage period with the microstructure of P(VDF–TrFE) molecular chain. Moreover, a theoretical model was constructed to describe the polarization fatigue evolution of P(VDF–TrFE) samples.

## 1. Introduction

In recent years, advancements in flexible electronics have enabled the fabrication of highly sensitive devices for wearable and portable devices. Recent efforts in the miniaturization of memory devices have contributed to increasingly flexible and portable devices [1,2,3,4,5]. Poly(vinylidene fluoride-trifluoroethylene) (P(VDF–TrFE)) have some outstanding advantages, such as high flexibility, high toughness, good workability, low cost, and so on. It has been widely applied in the ferroelectric random access memory (FeRAM), printed electronics, and wearable devices [6,7,8]. For low voltage operation of flexible devices, consuming less energy, it is essential to fabricate polymer films as thin as possible. The influencing factors of P(VDF–TrFE) ultrathin film were also reported, including temperature [7,9], materials of the electrode [10,11], thickness of films [12,13], switching cycles [14] and so on.

For an ultrathin film, such as 5060 nm, its ferroelectric properties are strongly dependent on the interface layers [15]. For instance, it was found that an electroactive interlayer [16,17] between electrode and P(VDF–TrFE) interlayer can improve the performance of polymer memory devices. The fatigue properties of P(VDF–TrFE) thin films under 10^6^–10^7^ switching cycles [18,19] have received considerable attention in recent years. As a form of fatigue, device stability [20,21] is defined as the loss of strength or other measure of performance as a result of the application or storage time under atmospheric conditions. The stability model for switching-induced charge-injection fatigue theory was proposed to explain the inherent fatigue mechanism of ferroelectric film [22]. As a result, temporal stability, the reliability of devices, can be used to estimate the properties of polymer devices for practical applications. 

In this work, different factors that may dominant the temporal stability of polymer devices were investigated. It was found that for P(VDF–TrFE) ultrathin films stored in air for 150 days, the value of $P_{r}/P_{r}(0)$ can keep at 0.80 at 25 °C and 0.70 at 60 °C, and the values of $P_{r}/P_{r}(0)$ obtained in our experiment are relative high compared with results from Zhu et al. [14]. Here, the polarization fatigue influence factors are investigated, including the testing temperature and the storage period; its influences on polarization properties of samples correlated with the microstructure of the P(VDF–TrFE) molecular chain and the HF formation inhibition mechanism. Meanwhile, the polarization fatigue was regulated by the switching model to achieve the temporal stability of the P(VDF-TrFE) ultrathin film capacitor. We systematically discuss the possible origins of these results in the context of designing the optimum protocol for full solution printed electronics based on P(VDF-TrFE) copolymer ultrathin films.

## 2. Materials and Methods

All capacitor samples were fabricated in the same manner. First, 3,4-ethylene dioxythiophene (EDOT) and poly(styrene sulfonic) acid (PSSH) with a mole ratio of 1:1.25 were freshly mixed and dissolved in deionized (DI) water at a concentration of 0.6%. Then, the mixture of poly(3,4–ethylene dioxythioohene)-poly(styrene sulfonic) acid (EDOT–PSSH) water solution was spin coated onto the wafer substrate which contained a thermal growth layer of 50 nm SiO_2_, and the wafer substrate was covered globally with titanium (Ti) as a bottom electrode. The EDOT–PSSH wafer was then heated up to 150 °C for 30 min to remove the absorbed water and gas on the surface of Si. After heating for 5 min on a hot plate, H_2_O_2_ (ω=2%) was spin coated on the EDOT–PSSH film to polymerize EDOT into PEDOT. After that, P(VDF–TrFE) (VDF/TrFE 70/30) copolymer solution was spin coated on the PEDOT–PSSH electroactive interlayer. After P(VDF–TrFE) was coated, the spin coating of PEDOT–PSSH was carried out to get the electroactive interlayer on the top of the P(VDF–TrFE) thin film. After that, all the samples were annealed at 130 °C for 1 h. Tens of Ti electrodes with diameters of 0.3 mm on top were prepared by the vacuum evaporation method through shadow masks.

Samples of fresh samples and samples after 7 days in storage were recorded as LC1 and LC2, respectively. Samples after 90 days and 150 days in storage were recorded as LC3 and LC4, respectively. For LC1 and LC2, the polarization fatigue of P(VDF–TrFE) ultrathin films were measured at 25 °C and 60 °C, respectively, and the pulse width of the applied voltage was 50 μs. For LC3 and LC4, the pulse width of the applied voltage was 30 μs. The polarization fatigue of P(VDF–TrFE) ultrathin films were measured with a Precision Pro Ferroelectric Tester (Radiant Technologies, Inc, Albuquerque, NM, USA). A 10 Hz triangular-wave electric field was used to obtain the P–E hysteresis loops of the films. The thicknesses of thin films were measured by a surface profiler (SCIENTECH, Taipei, Taiwan). All samples were measured on a constant temperature heating table (HP–1010), and samples were fatigued by applying a bipolar rectangular electric field of 200 MV/m and frequency of 10 Hz. The permittivity of all samples was 15.

## 3. Results and Discussion

### 3.1. Temporal Stability under the Influence of Multiple Factors

In order to investigate the temporal stability of P(VDF–TrFE) ultrathin film capacitors, the $P_{r}/P_{r}(0)$ of LC1–LC4 samples were measured at 25 °C and 60 °C. As is well known, with the same number of switching cycles, the value of $P_{r}/P_{r}(0)$ will decrease with the increase of voltage pulse width [23], since more and more defects will be produced internally [24] with longer voltage action time in a single switching cycle. The defects will inhibit the nucleation of reverse ferroelectric domains, thus preventing the reversal of ferroelectric domains and causing fatigue of the sample. As a result, the fatigue process of polarization switching can be accelerated, that is, the longer the voltage acting time, the faster the fatigue process is. Therefore, considering the influence of storage periods and applied voltage pulse width, we apply 50 μs pulsed voltage for LC1 and LC2 to accelerate the fatigue process, and 30 μs pulsed voltage for LC3 and LC4 to study the fatigue process in more detail. However, based on our recent findings, for pulse widths of 30 μs and 50 μs, which are relatively close to each other (differed by less than 67%), the polarization performance of our test samples differed by less than 20% under the same conditions. Research about the effect of pulse width on fatigue of samples are currently ongoing.

Figure 1 shows the $P_{r}/P_{r}(0)$ dependence of the switching cycles for four groups of capacitors. As can be seen in Figure 1, at the beginning, different storage periods and testing temperatures have a slight influence on $P_{r}/P_{r}(0)$. However, after 10^5^ cycles, it was obvious that the value of $P_{r}/P_{r}(0)$ decreased rapidly with increased storage period and switching cycles, and the process sped up at 60 °C. However, after 10^6^ cycles, the relative remanent polarization of LC4 (stored in air for 150 days) can still be maintained at a relatively high level [14] (0.80 at 25 °C and 0.70 at 60 °C), which demonstrates that the samples can still maintain good temporal stability even after long-term storage.

To further explore this result, a series of experiments were carried out, and the polarization properties of LC1–LC4 with different storage periods and testing temperatures are listed in Table 1. From Table 1, we can see that the initial remanent polarization ($P_{r}(\mathrm{BF})\;$) of the sample decreased with the increase of storage period, and after being stored in air for 150 days, $P_{r}(\mathrm{BF})\;$ decreased by 24.4% both at 25 °C and 60 °C. Meanwhile, the relative remanent polarization reflects the polarization characteristics of ferroelectric thin film devices in the polarization switching process. Therefore, we pay more attention to the characteristics of relative remanent polarization in the process of polarization switching for the stability of samples.

Figure 2 shows the P–E hysteresis loops of LC1–LC4 measured at 25 °C. The solid lines in black denote the hysteresis loops before fatigue and the solid lines in red denote the hysteresis loops after fatigue. It was found that the value of $E_{c}$ keeps at a relatively stable level from 62 to 72 MV/m both at 25 °C and 60 °C, which implies that the coercive field of samples show excellent temporal stability. In addition, the value of $P_{r}$ reduced by 14.3% (from 7.0 μC/cm^2^ to 6.0 μC/cm^2^) at 25 °C, while the value of $P_{r}$ reduced by 24.6% (from 6.9 μC/cm^2^ to 5.2 μC/cm^2^) at 60 °C, indicating that an increase in temperature accelerated the degradation of the polarization properties. This can be associated with trapped charges, as trapped charges will increase with increasing testing temperatures [25], resulting in a decline of the temporal stability of the device.

### 3.2. The Trapped Charge Model of Polarization Switching Regulation

The relationship between the trapped charge density ($\rho_{trap}$) and the remanent polarization ($P_{r}$) can be described by [26]:(1)$$P_{r}=\frac{3\varepsilon_{r}+3}{4\varepsilon_{r}+1}(\varepsilon_{r}\varepsilon_{0}E_{0}-q\rho_{trap}W)$$
where $\varepsilon_{0}$ is the permittivity of vacuum, $\varepsilon_{r}$ is the permittivity of ferroelectric films, $E_{0}$ is the applied electric field, q is the electron charge, and W is the depletion width.

To investigate the effects of trapped charge density on polarization properties, the polarization fatigue properties of LC1–LC4 were obtained at 25 °C, as shown in Figure 3. Apparently, the remanent polarization ($P_{r}$) declined as the trapped charge density ($\rho_{trap}$) increased. In addition, during the entire polarization switching process, the average remanent polarization ($\bar{P_{r}}$) of samples LC1, LC2, LC3, and LC4 were 9.00, 8.90, 6.72, and 6.66 μC/cm^2^, respectively. According to Equation (1), the $\rho_{trap}$ values of sample LC1, LC2, LC3 and LC4 were 7.2 × 10^23^ m^−3^, 9.0 × 10^23^ m^−3^, 4.7 × 10^24^ m^−3^, and 4.8 × 10^24^ m^−3^, respectively.

Then, the quantitative relationship between the remanent polarization of the capacitor sample and the trapped charge density was obtained. Comparing the experimental results of LC1 with LC2, we found that $P_{r}$ decreased by 1.0 × 10^−^^3^ C/m^2^. Meanwhile, we can deduce from Equation (1) that $\rho_{trap}$ was increased by 1.8×10^23^/m^3^. Comparing LC3 with LC4, when $P_{r}$ decreased by 6.0 × 10^−4^ C/m^2^, $\rho_{trap}$ value increased by 1.0 × 10^23^/m^3^, as listed in Table 2. Therefore, the quantitative relationship between trapped charge density and remanent polarization can be expressed as $\Delta P_{r}=-5\times 10^{-27}\cdot \Delta \rho_{trap}+1\times 10^{-27}$.

### 3.3. The Electroactive Interlayer Inhibits HF Elimination

External factors, such as ultraviolet irradiation [27], water vapor [28], and environmental humidity [29], can affect the performance of polarization. In order to investigate the theoretical model of time stability for capacitors, it is essential to explore the device performance and the internal microstructure of ferroelectric ultrathin films. As shown in Figure 4, the adjacent hydrogen and fluorine atoms in the molecule chain can combine to form HF under the influence of external factors [30,31]. It is believed that the occurrence of dehydrofluorination is mainly induced by trapped charges during switching cycles [32], and this will result in the degradation of the polarization properties of P(VDF-TrFE) [33].

Figure 5 shows the polarization state of P(VDF–TrFE) thin films in ferroelectric capacitors. Escape of HF from the molecular chain will result in the formation of a conjugated polyene sequence and crosslinked polymers [34]. From Figure 5, we can see that the interlayer PEDOT–PSSH supplies charges needed for the compensation of dipoles to stabilize the active domain. Hence, PEDOT–PSSH can inhibit the occurrence of bond breaking, and thus inhibit the formation of HF to some extent.

In addition, it was found that the electroactive interlayer acts as a protective layer to reduce the adverse effect of the external environment on the P(VDF–TrFE) sample, since multiple environmental factors, such as temperature, humidity, long-term storage, etc. [35,36], can cause a series of changes on the surface morphology and crystallinity of P(VDF–TrFE) films. Furthermore, an electroactive interlayer can improve the crystallinity of P(VDF–TrFE) films [37]. Therefore, the polarization properties of P(VDF–TrFE) sample can be stabilized at a relatively high level after being stored in air for several months and exhibit high temporal stability.

## 4. Conclusions

The polarization properties of the poly (vinylidene fluoride-trifluoroethylene) [P(VDF-TrFE)] ultrathin film devices with testing temperature and storage period were studied. By comparing experimental results with the theorical model, the linear relationship between trapped charge density and remanent polarization was obtained. In addition, it was also found that an interlayer can reduce the trapped charge density and inhibits the elimination of HF from the molecular chain, thus improving the polarization properties of the sample. This model provides ideas for designing optimization schemes based on P(VDF-TrFE) copolymer ultrathin film devices, and is an important reference for improving and regulating multi-time storage and high efficiency devices.

## Figures and Tables

**Figure 1 membranes-11-00301-f001:**
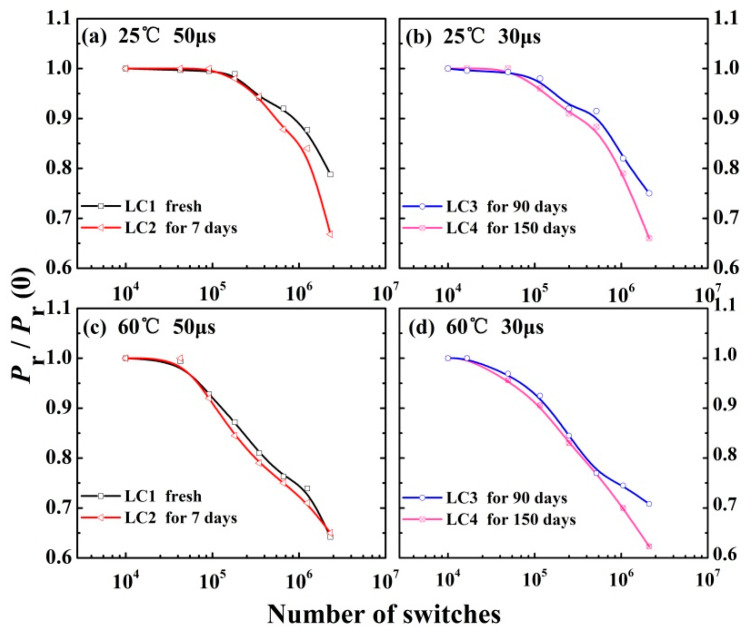
The polarization properties of P(VDF–TrFE) films stored in air for several days with 50 μs applied voltage pulse width at 25 °C (**a**) and 60 °C (**c**). The polarization properties of P(VDF–TrFE) films stored in air for 90 or 150 days with 30 μs applied voltage pulse width at 25 °C (**b**) and 60 °C (**d**).

**Figure 2 membranes-11-00301-f002:**
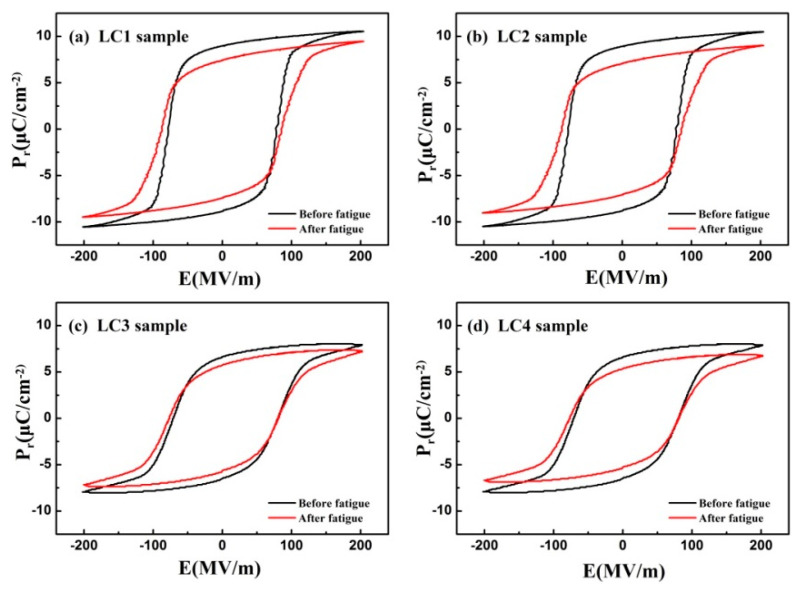
The P–E hysteresis loops of LC1–LC4 measured at 25 °C. The solid lines in black denote the hysteresis loops before fatigue, and the solid lines in red denote the hysteresis loops after fatigue.

**Figure 3 membranes-11-00301-f003:**
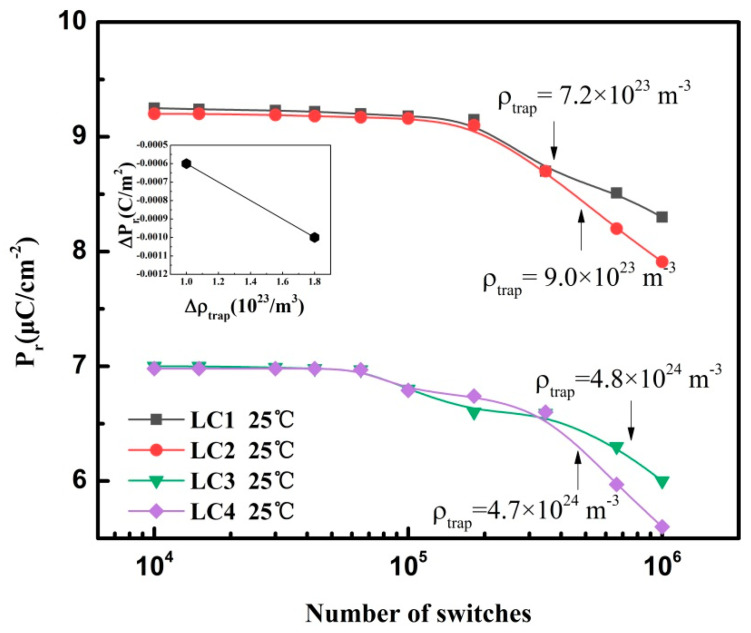
Polarization fatigue properties of LC1–LC4 with different trapped charge densities at 25 °C. The inset in the top left corner shows the linear relationship between trapped charge density and remanent polarization.

**Figure 4 membranes-11-00301-f004:**
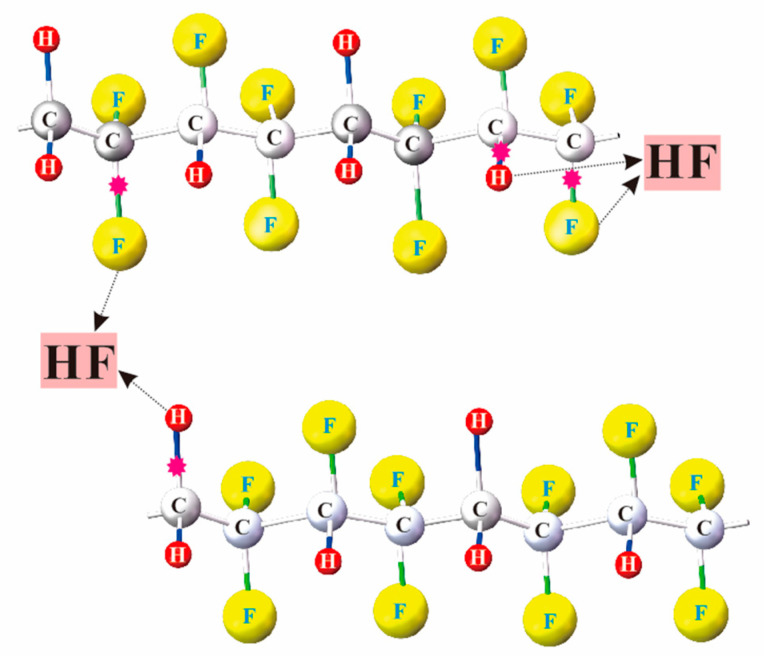
The all-trans molecular conformation of P(VDF–TrFE). The black dotted line indicated that the hydrogen atom (H) and the fluorine atom (F) can combine to form hydrofluoric acid (HF).

**Figure 5 membranes-11-00301-f005:**
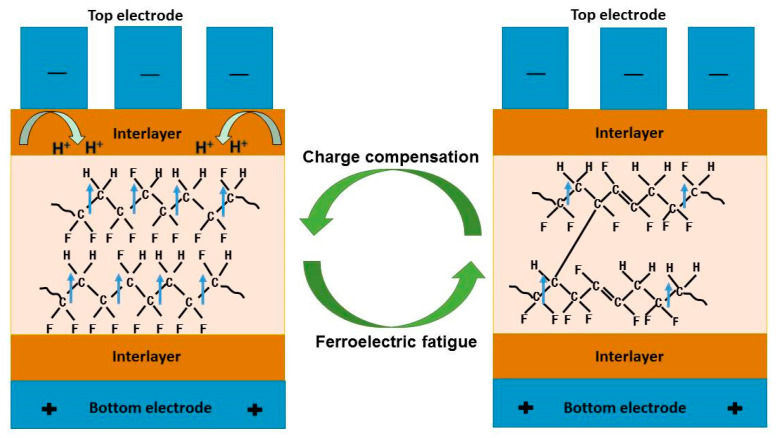
Schematic diagram of the polarization state of P(VDF–TrFE) thin films in ferroelectric capacitors, including the processes of charge compensation and ferroelectric fatigue.

**Table 1 membranes-11-00301-t001:** The remnant polarization ($P_{r}$) and coercive field ($E_{c}$) for the LC1–LC4 samples before and after fatigue at 25 °C and 60 °C.

		LC1	LC2	LC3	LC4
		Fresh	7 Days	90 Days	150 Days
25 °C	$P_{r}({\mathrm{BF})\;\mu C/\mathrm{cm}}^{2}$	9.25	9.20	7.00	6.98
	$P_{r}({\mathrm{AF})\;\mu C/\mathrm{cm}}^{2}$	8.30	7.91	6.00	5.60
	Ratio(AF/BF)	0.90	0.86	0.85	0.80
	Ec(BF) MV/m	64.5	64.1	64.3	63.5
	Ec(AF) MV/m	71.4	70.5	65.7	65.1
60 °C	$P_{r}({\mathrm{BF})\;\mu C/\mathrm{cm}}^{2}$	9.05	9.02	6.90	6.85
	$P_{r}({\mathrm{AF})\;\mu C/\mathrm{cm}}^{2}$	6.80	6.40	5.20	4.80
	Ratio(AF/BF)	0.75	0.71	0.75	0.70
	Ec(BF) MV/m	64.6	63.7	62.1	61.9
	Ec(AF) MV/m	71.1	70.9	65.7	64.9

**Table 2 membranes-11-00301-t002:** The remnant polarization ($P_{r}$), the change values of the remnant polarization ($\Delta P_{r}$), and the change values of trapped charge density ($\Delta \rho_{trap}$) for the LC1–LC4 samples.

Samples	LC1	LC2	LC3	LC4
$P_{r}/\;C\cdot m^{-2}$	9.00 × 10^−2^	8.90 × 10^−2^	6.72 × 10^−2^	6.66 × 10^−2^
$\Delta P_{r}/\;C\cdot m^{-2}$	1.0 × 10^−3^		6.0 × 10^−4^	
$\Delta \rho_{trap}/\;m^{3}$	1.8 × 10^23^		1.0 × 10^23^	

## Data Availability

Data sharing not applicable. No new data were created or analyzed in this study.

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
