# Peer review of "The Influence Mechanism of Temperature and Storage Period on Polarization Properties of Poly (Vinylidene Fluoride–Trifluoroethylene) Ultrathin Films"

_membranes, 2021, doi:10.3390/membranes11050301_

Round 1

Reviewer 1 Report

In this communication, the authors reported the polarization properties of P(VDF-TrFE) ultrathin films stored for different periods of time and tested at two temperatures. Based on their observations, they developed a quantitative relationship between trapped charge density vs remanent polarization of the samples, and proposed a theoretical model describing the escape of HF to interpret the results.

This manuscript is concise and in general decently-written, but the authors need to check carefully on their grammar and spelling, as there are quite a few mistakes that are hard to ignore. The resolution of figures should also be improved and the format of citation should be consistent. A few questions/comments regarding the content:

  • The authors stated that after being stored for 5 months, the relative remanent polarization of the material is still high – this statement needs to be justified. Is the relative remanent polarization high compared to values reported in other papers? Are the testing conditions comparable?
  • The initial remament polarization of LC4 is significantly lower compare to fresh samples – what does this low value indicate in terms of the temporal stability of LC4? Looks like the samples already aged a lot before being tested.
  • Samples were tested with different pulse width, and the authors did explain that higher pulse width can accelerate the fatigue process. Therefore, was the high level of relative remanent polarization of LC3/LC4 due to the lower pulse width? How different would the results be if they were tested with the higher pulse width?
  • In section 3.2, it would be helpful to summarized the numbers in a table. It’s also unclear how the relationship was developed. How was the factor 5*10^-27 calculated using the values 0.097, 1.8*10^23, 0.064 and 1*10^23? Delta Pr should be negative since remanent polarization goes down.

Some line-specific comments:

  • Line 94: Why applying a lower pulse width would allow the authors to study the fatigue process in more detail? Wouldn’t the trend be the same?
  • Line 100: Typo “remnant” polarization
  • Line 110 & line 115: The authors stated that the samples have good or excellent temporal stability. Please address the questions above on the initial low remanent polarization and low pulse width used for the samples.
  • Line 137: The _ values?

Reviewer 2 Report

Li et, al. reported their study on the influence mechanism of temperature and storage period on the polarization of P(VDF-TrFE). The trapped charge model is introduced and the degradation mechanism is analyzed. The provided information could be interesting for the researchers in related fields, but these questions should be properly addressed before publication.

  1. How is the trapped charge density measured? Is it a measured value or a calculated value?
  2. Is there any evidence or initial experimental validation to prove the hypothesis in section 3.3?
  3. How can the model correlate to the experimental observation found in the first section?

Reviewer 3 Report

This paper is on the influence of the mechanism of temperature and storage period on polarization properties of a ferroelectric polymer. The study is not well presented and I have some remarks and questions to the authors.

In the introduction, lines 49 to 59, the study is very detailed (too detailed in my mind) but all points are not included in the study…

The first question is why did you chose fresh/ 7 days / 3 months and 5 months stored time? A regular storage period between the samples would be more interesting to study the storage period. Then, it is preferable to have the same scale, in days for example (3 months correspond to 90 days or 91, 92?). In fact, no explanation has been done in the study concerning the difference in polarization between a fresh and an old sample. Even if, there is 4 samples, no comparison between them has been really realized. The authors explain only fatigue process.

The methods are not very clear. How did you measure at 25°C and 60 °C? Is there 8 samples? How many electrodes do you have on each sample? Is the measure has been done at 25°C and then at 60°C?

The pulse voltage is not define. Is it a rectangular pulse? What is its amplitude? Is it positive and negative? Why did you accelerate the fatigue process? I don’t understand why 30 µs permits you to study the fatigue process in more detail. Change the fatigue process at 50 µs to 30 µs seems to be not scientific.

It would be interesting to show the initial P-E loop of each sample and a P-E loop after fatigue. As the Ec values in table 1 are high for LC3 comparing to a classic ferroelectric material, the form of the P-E loop will be interesting to know if you have a hard or a soft ferroelectric and if there is some dielectric losses.

Why is there no value of Ec for LC1, LC2 and LC4? The study seems to be not complete…

Table 1: The number of switches is not mentioned for the values after fatigue. What is the error on the measure? The evolution of the ratio (AF/BF) between the sample at 25°C and 60°C is not explained. Why LC3 has a better ratio than LC2 at 60°C? It will be interesting to have a sample with a larger storage period (1 year for example). For all values, it will be also interesting to compare with the values of the literature.

For the trapped charge model, what is the value of the permittivity of ferroelectric films? Is it the same value for all samples?

The insert in the figure 2 is not clear. The straight line seems to correspond to the equation at the line 145 p5 but no experimental point is visible.

The sentence “Apparently, the remanent polarization Pr declined as the trapped charge density rtrap increased” is an evidence if you considered the equation (1)… What is the trapped charge in your samples ?

Page 5, line 150: Please explain the “external factors”.

Page 5, line 160: the reference will be in brackets.

The part 3.3 explains a model of the electroactive interlayer that inhibit HF elimination and in the conclusion it is mentioned that “In addition, it is also found that an interlayer can reduce the trapped charge density and inhibits the elimination of HF” but no experiment confirms this theory in the study. The theory and the experiments are not really correlated.

Round 2

Reviewer 1 Report

The authors have addressed the comments in the response letter, but did not add relevant references and discussions (included below) to their manuscript. Such information should be provided to readers so that they can have a better understanding of the work.

Starting line 91: "As is well known, with the same number of switching cycles, the value of P P/ (0) r r will decrease with the increase of voltage pulse width, and the fatigue process of polarization switching can be accelerated, that is, the longer voltage acting time the faster the fatigue process is." While in the response letter: "Based on our recent findings, for the pulse width of 30 μs and 50 μs which are relatively close to each other, its effect on the polarization performance is negligible. Therefore, this paper only sets two different pulse widths as controls, which will not affect the experimental results obtained." The statements are inconsistent, please clarify. 

Comment 1: The authors stated that after being stored for 5 months, the relative remanent polarization of the material is still high – this statement needs to be justified. Is the relative remanent polarization high compared to values reported in other papers? Are the testing conditions comparable?

Answer: According to the experimental results reported by Zhu et al [1], after 106 switching cycles, the relative remanent polarization of fresh P(VDF-TrFE) is decreased to about 60% and 55% with testing temperature of 30℃ and 70℃ respectively. While, in our study, after 106 switching cycles, the relative remanent polarization of LC4 (after being stored in air for 150 days) is decreased to 80% and 70% with testing temperature of 25℃ and 60℃ respectively, the testing conditions were similar, therefore, we can state that after being stored in air for 150 days, the relative remanent polarization of the material is still high.

 [1] G.D. Zhu, X.Y. Luo, J.H. Zhang and Y. Gu. Electrical fatigue in ferroelectric P(VDF-TrFE) copolymer films[J]. IEEE Transactions on Dielectrics and Electrical Insulation, 2010, 17, 1172-1177.

Comment 7:        Line 110 & line 115: The authors stated that the samples have good or excellent temporal stability. Please address the questions above on the initial low remanent polarization and low pulse width used for the samples.

Answer: As for the initial low remanent polarization, the polarization performance of the sample decreases with the increase of storage period, therefore, the initial residual polarization intensity of the sample decreases after a long storage period. Meanwhile, the relative remanent polarization reflects the polarization characteristics of ferroelectric thin film devices in the polarization switching process. Therefore, we pay more attention to the characteristics of relative remanent polarization in the process of polarization switching in order to manifests the stability of samples. The high values of the measured remanent polarization under multi-time storage conditions prove the stability achieved by the device.

Based on our recent findings, for the pulse width of 30 μs and 50 μs which are relatively close to each other, its effect on the polarization performance is negligible. Therefore, this paper only sets two different pulse widths as controls, which will not affect the experimental results obtained.

Reviewer 3 Report

The authors have addressed the comments in the response letter. Therefore, the manuscript would be improved if they add discussions included in the response letter to their manuscript. Such information should be provided to readers so that they can have a better understanding of the work (For example the permittivity or the external factors are not mentionned).

It would be also preferable to mention the electric field applied and not the voltage (line 86 p2) and P-E loops and not P-V loops in figure 2.

The reviewer comments are for improved a poor publication and not only to be answer in a private letter.

Round 3

Reviewer 1 Report

Comments have been addressed by the authors.
